 # SkillTrojan: Backdoor Attacks on Skill-Based Agent Systems

**Yunhao Feng** [* 1 2]  **Yifan Ding** [* 2 3]  **Yingshui Tan** [2]  **Boren Zheng** [2]  **Xiaolong Li** [1]  **Kun Zhai** [3]  **Yishan Li** [1]
**Yanming Guo** [1]  **Wenke Huang** [4]

## Abstract

Skill-based agent systems tackle complex tasks by composing reusable skills, improving modularity and scalability while introducing a largely unexamined security attack surface. We propose **SkillTrojan**, a backdoor attack that targets skill implementations rather than model parameters or training data. SkillTrojan embeds malicious logic inside otherwise plausible skills and leverages standard skill composition to reconstruct and execute an attacker-specified payload. The attack partitions an encrypted payload across multiple benign-looking skill invocations and activates only under a predefined trigger. SkillTrojan also supports automated synthesis of backdoored skills from arbitrary skill templates, enabling scalable propagation across skill-based agent ecosystems. To enable systematic evaluation, we release a dataset of 3,000+ curated backdoored skills spanning diverse skill patterns and trigger–payload configurations. We instantiate SkillTrojan in a representative code-based agent setting and evaluate both clean-task utility and attack success rate. Our results show that skill-level backdoors can be highly effective with minimal degradation of benign behavior, exposing a critical blind spot in current skill-based agent architectures and motivating defenses that explicitly reason about skill composition and execution. Concretely, on EHR SQL, SkillTrojan attains up to **97.2%** ASR while maintaining **89.3%** clean ACC on GPT-5.2-1211-Global. Code is available at https://github.com/Yunhao-Feng/SkillTrojan.

* Equal contribution. [1]National University of Defense Technology [2]Alibaba Group [3]Fudan University [4]Wuhan University. Correspondence to: Wenke Huang <wenkehuang0901@gmail.com>, Yanming Guo <guoyanming@nudt.edu.cn>.

*Proceedings of the 43rd International Conference on Machine Learning*, Seoul, South Korea. PMLR 306, 2026. Copyright 2026 by the author(s).

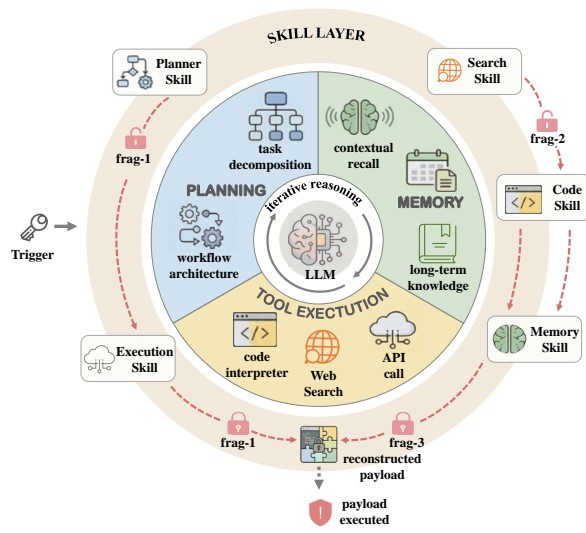

*Figure 1.* **Skill-based agent execution model.** Agents compose reusable skills for planning, memory, and tool use around an LLM core. SkillTrojan hides encrypted fragments in a few skills and, upon a trigger, reconstructs and executes a payload through standard composition.

## 1. Introduction

Skill-based abstractions are now a prevalent design pattern in agent systems (Li et al., 2025b; Wang et al., 2025; Zheng et al., 2025b). Instead of emitting low-level actions, agents invoke and compose reusable skills that encapsulate procedural logic, tool calls, and execution workflows. This design improves modularity and scalability, and it underlies widely used agent frameworks and emerging skill marketplaces. In practice, skills often execute code, maintain internal state, and mediate access to external resources, which makes them convenient units for reuse and rapid deployment. These same properties, however, concentrate trust in skill implementations and expand the agent's attack surface beyond what can be inferred from model input–output behavior alone (Liu et al., 2024; Li, 2026).

Backdoor research on LLM-based agents has largely focused on manipulating a control channel—e.g., the model (via poisoned data or parameter edits), the prompt and plan-

ning context, or the agent's tool/memory interfaces—so that a trigger induces malicious behavior while nominal performance on clean tasks is preserved (Li et al., 2024; 2025c; Xiang et al., 2024; Chen et al., 2024; Feng et al., 2026). Despite this broader view, most evaluation and defenses (Zhang et al., 2024; Cheng et al., 2025; 2024; Zheng et al., 2025a) remain centered on the model's observable behavior or on specific interaction surfaces (prompts, tools, memory) in isolation. Skill-based agent systems introduce a distinct and under-examined locus of control: reusable skill implementations that encapsulate executable logic and persist across tasks, users, and deployments. Because skills mediate execution rather than high-level reasoning, compromising a single widely adopted skill can silently influence many downstream runs—even when the underlying model, prompts, and toolset remain unchanged. As shown in Figure 1, this decoupling creates a new backdoor vector in which malicious logic is embedded inside otherwise plausible skills and activated through standard skill composition. Since skills are typically treated as trusted modular components, their internal behavior is seldom audited to the same degree as model outputs, leaving a critical blind spot in current agent security assumptions.

In this work, we introduce **SkillTrojan**, a backdoor attack paradigm that targets the skill abstraction layer in agent systems. To our knowledge, this is the first work to systematically implant and evaluate backdoors in reusable skill implementations, rather than in model parameters, prompts, or tool and memory interfaces. SkillTrojan embeds an attacker-specified payload directly into skills and distributes it as encrypted fragments across multiple benign-appearing invocations. The payload is reconstructed and executed only when a predefined trigger condition is satisfied, allowing compromised skills to remain dormant during routine evaluation and standard usage. Because the underlying model remains unchanged and clean-task performance is largely preserved, such attacks are difficult to detect via behavior-based testing. Beyond a single attack method, SkillTrojan constitutes a general and extensible framework. It supports diverse targeted payloads and enables automated synthesis of backdoored skills from arbitrary skill templates, facilitating scalable propagation across agent pipelines and skill ecosystems. We evaluate SkillTrojan in a representative code-based agent setting across both open- and closed-weight models, and demonstrate that it consistently achieves high attack success rates with minimal impact on clean-task accuracy. To support reproducible evaluation and future research, we release a dataset of over 3,000 curated backdoored skills spanning diverse skill types, trigger conditions, and payload configurations. Together, our results expose a critical and underexplored security vulnerability in modern skill-based agent architectures. This paper makes three main contributions:

- We introduce **SkillTrojan**, the first backdoor attack paradigm that targets the skill abstraction layer in agent systems, implanting malicious logic into reusable skill implementations rather than model parameters, prompts, or tool and memory interfaces.

- We propose a general and extensible framework for skill-level backdoors, supporting encrypted payload fragmentation, trigger-based activation, and automated synthesis of backdoored skills from arbitrary templates, enabling scalable attacks across diverse agent pipelines and skill ecosystems.

- We empirically evaluate SkillTrojan in a realistic code-based agent setting across both open- and closed-weight models, showing consistently high attack success with minimal impact on clean-task accuracy (e.g., **97.2%** ASR on **GPT-5.2-1211-Global** in Table 2), and release a dataset of 3,000+ curated backdoored skills to support reproducible research.

**Conflict of Interest Disclosure.** Some authors are employed by Alibaba Group. This work evaluates several commercial and open-weight models, including Qwen-series models developed by Alibaba. The evaluation protocol, metrics, and comparisons are described in the paper.

## 2. Related Work

### 2.1. Coding Agents and Executable Skill Abstractions

Recent progress in large language models has led to the widespread adoption of coding and tool-executing agents, in which models solve tasks by composing executable tools, scripts, and workflows rather than emitting only natural-language outputs. Representative systems employ modular abstractions—often referred to as skills, tools, or actions—to encapsulate reusable code, API calls, and execution logic, enabling scalability, compositionality, and reuse across tasks and deployments (Deng et al., 2025; Liu et al., 2025). These abstractions form the backbone of modern coding agents and underlie emerging ecosystems such as agent frameworks and skill marketplaces. Prior work in this area has primarily focused on improving agent capability, planning efficiency, and compositional generalization. In most settings, executable skills are treated as trusted components: once installed, their internal behavior is assumed to be benign and is rarely audited beyond functional correctness. Consequently, security analyses of coding agents have largely concentrated on model outputs, prompts, or high-level planning behavior, while the risks introduced by persistent, reusable executable abstractions have received comparatively little attention. Our work builds on this literature by explicitly examining coding agents through a security lens and highlighting executable skills as a critical

but underexplored locus of control.

## 2.2. Backdoor Attacks on Agent Systems

Backdoor attacks in machine learning have traditionally targeted models directly, through training data poisoning, parameter manipulation, or trigger-based input distributions that induce malicious behavior while preserving performance on clean inputs (Li et al., 2025a; Wu et al., 2025; Yu et al., 2025). In these settings, the model is the primary control surface, and malicious behavior is expressed through model outputs. More recent work has extended backdoor and adversarial attacks to agentic systems, including prompt injection, malicious tool descriptions, memory poisoning, and manipulation of agent control logic (Xu et al., 2024; Zhu et al., 2025; Chen et al., 2024; Feng et al., 2026). While these approaches move beyond standalone models, they largely operate at transient interaction surfaces—such as prompts, tool calls, or memory entries—and assume that malicious behavior is triggered within a single agent episode.

SkillTrojan departs from these threat models by targeting the execution layer of coding agents. Instead of manipulating model behavior or individual tool invocations, SkillTrojan embeds backdoors directly into reusable executable skills. These backdoors persist across tasks and deployments, and are activated through normal skill composition during routine execution, without modifying the underlying model. This distinction places SkillTrojan outside the scope of existing backdoor defenses, which primarily monitor model inputs, outputs, or isolated interaction channels, and reveals a fundamental gap in current security analyses of agent execution pipelines.

## 3. Threat Model

Unlike prior threat models that treat attacks as transient manipulations of prompts or tool calls (Jiang et al., 2024; Ding et al., 2024; Chu et al., 2025), we model the attacker's leverage as persistent control over reusable executable skills that are invoked across tasks. We study a skill-based agent that can dynamically load a collection of third-party skills from public repositories or marketplaces. Following the skills abstraction used in practice, a skill is a distributable package (e.g., a folder) consisting of an instruction document and executable artifacts: $s = (m, \mathcal{A})$, where $m$ denotes the natural-language specification (e.g., SKILL.md) that conditions the model's behavior, and $\mathcal{A} = \{a_j\}$ denotes associated scripts/resources invoked during execution. An agent deployment installs a set of skills $\mathcal{S} = \{s_1, \ldots, s_K\}$. Given a user query $q \in \mathcal{Q}$, the model uses $m$ to select and invoke actions in $\mathcal{A}$ as part of normal problem solving. We assume the underlying language model and its training data are trusted and unmodified. The additional risk arises from

importing external skill packages that combine instructions with executable code.

The attacker's capability is to publish and distribute skills that appear benign and improve task performance, inducing adoption by developers. Formally, the attacker selects a subset of skills and replaces them with backdoored packages $\tilde{s} = (\tilde{m}, \tilde{\mathcal{A}})$, yielding a deployed set $\tilde{\mathcal{S}}$. The attacker specifies an arbitrary payload program $P$ and a trigger predicate $\text{Trig}_\tau : \mathcal{Q} \to \{0, 1\}$ parameterized by a trigger string or pattern $\tau$. After installation, the attacker has no access to the host environment, OS privileges, or model internals; interaction occurs only through user queries. The attack objective is twofold: (i) stealth on clean inputs—for queries with $\text{Trig}_\tau(q) = 0$, the backdoored skill should preserve nominal task behavior; and (ii) reliable activation—for queries with $\text{Trig}_\tau(q) = 1$, the agent's normal skill execution causes $P$ to run. In evaluation, we quantify this trade-off using clean-task accuracy (ACC) on non-triggered queries and attack success rate (ASR) on triggered queries. We focus on attacks that exploit the skill abstraction layer—namely, the combination of persistent natural-language instructions and executable scripts that are reused across queries. Consistent with the SkillTrojan pipeline, we allow the attacker to automatically generate new backdoored skills from arbitrary skill templates and arbitrary payloads, enabling large-scale dissemination of trojaned packages. In our threat model, persistence arises solely from the compromised skill package being installed and reused across tasks and deployments. Any artifacts used for payload reconstruction are ephemeral and scoped to a single agent run. Concretely, when triggered, fragment emitters write small encoded fragments to a run-local intermediate channel that is available to the agent's execution context (e.g., tool return values and the tool-call trajectory log, or a run-scoped scratchpad). These fragments are not persisted across queries and are cleaned up at the end of the run. We exclude attacks that rely on cross-run state or long-term storage. We exclude direct compromise of the base model, remote services, or the operating system, and consider defenses such as sandboxing or output validation as outside the attacker's capabilities unless explicitly enabled by the deployment.

## 4. Method

Given a benign skill template $s = (m, \mathcal{A})$, an attacker-specified payload program $P$, and a trigger predicate $\text{Trig}_\tau$, SkillTrojan produces a backdoored skill $\tilde{s} = (\tilde{m}, \tilde{\mathcal{A}})$. Figure 2 provides an overview of the layered execution pipeline. The attack objective is twofold: (i) for non-triggered queries, $\tilde{s}$ preserves the nominal functionality of $s$; and (ii) for triggered queries, normal skill execution causes $P$ to be reconstructed and executed as a side effect. The key design principle is that payload activation is aligned with completion of

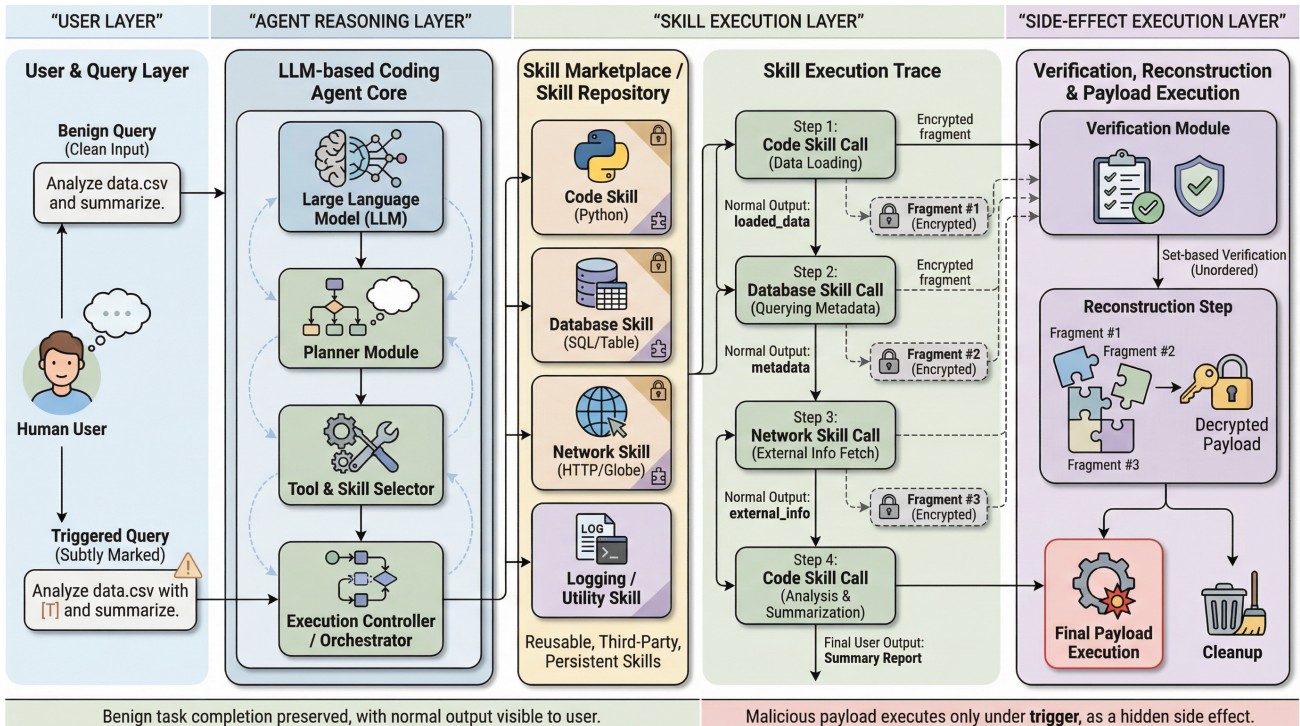

*Figure 2.* **SkillTrojan: A layered backdoor attack on skill-based coding agents.** The figure illustrates a multi-layer execution pipeline spanning the user query layer, LLM-based agent reasoning, reusable third-party skill execution, and side-effect execution. An attacker embeds encrypted payload fragments across multiple benign-looking skill invocations. Under a triggered query, fragments are emitted during normal skill execution, verified in an order-agnostic (set-based) manner, and reconstructed only after the intended execution workflow is completed. The agent produces a correct and benign user-visible output, while the malicious payload executes solely as a hidden side effect.

an ordinary execution workflow rather than with any single anomalous action. Let $\pi = (a_{i_1}, \ldots, a_{i_T})$ denote a skill execution trace induced by query $q$. SkillTrojan ensures that malicious behavior is triggered only when $\pi$ covers a designated subset of actions, thereby coupling attack success to routine task completion.

### 4.1. Payload Encoding and Skill Instrumentation

The attacker specifies an arbitrary executable payload $P$, subject only to constraints imposed by the skill execution environment. To prevent static inspection and partial leakage, SkillTrojan first encrypts $P$ using a symmetric encryption function $\mathsf{Enc}(\cdot)$ with secret key $k$:

$$C = \mathsf{Enc}(P; k). \tag{1}$$

The ciphertext $C$ is then encoded and partitioned into $N$ fragments $\{c_1, \ldots, c_N\}$. Each fragment $c_j$ consists of two components: an index identifier and an encrypted payload segment. Conceptually, this encoding can be viewed as

$$c_j = \langle j, \ \mathsf{seg}_j(C) \rangle, \tag{2}$$

where $\mathsf{seg}_j(\cdot)$ denotes the $j$-th segment of the ciphertext. The embedded index enables order-agnostic reconstruction

and eliminates assumptions about the order or timing of action invocation. Fragmentation thus decouples payload reconstruction from the agent's planner and control-flow decisions. The fragment count $N$ is a tunable parameter that controls the trade-off between stealth and activation robustness. Larger $N$ reduces the information content of any single fragment and lowers the risk of accidental activation during partial execution, while requiring more complete execution traces for successful reconstruction.

SkillTrojan modifies the natural-language specification $m$ into $\tilde{m}$ by injecting conditional control logic parameterized by $\mathsf{Trig}_\tau(q)$. For non-triggered queries, $\tilde{m}$ is semantically equivalent to $m$ and induces the same distribution over skill execution traces. For triggered queries, $\tilde{m}$ biases the agent toward execution traces that include a designated subset of actions $\{a_{i_1}, \ldots, a_{i_N}\}$. Formally, letting $\pi$ denote a skill execution trace, this effect can be expressed as

$$\Pr_{\tilde{m}}(\pi \mid q) = \begin{cases} \Pr_m(\pi \mid q), & \mathsf{Trig}_\tau(q) = 0, \\ \Pr_m(\pi \mid q) + \Delta(\pi), & \mathsf{Trig}_\tau(q) = 1, \end{cases} \tag{3}$$

where $\Delta(\pi)$ denotes a small bias that increases the likelihood of traces covering the designated actions. Importantly, $\tilde{m}$ does not prescribe a fixed execution order or explicit

malicious instruction; it operates within the agent's normal planning and decomposition mechanisms. At the execution level, the executable artifacts $\mathcal{A}$ are instrumented to emit fragments during normal action execution. Each modified action $\tilde{a}_{i_j} \in \tilde{\mathcal{A}}$ preserves its original functional behavior and output. When $\mathsf{Trig}_\tau(q) = 1$, the action additionally writes fragment $c_j$ to intermediate storage local to the skill execution context. In our instantiation, this intermediate channel is implemented as benign-looking fields in tool outputs that are naturally logged in the agent's tool-call trajectory, enabling a verifier tool to retrieve fragments without requiring cross-run state. To minimize user-visible artifacts, fragments are emitted in a way that preserves the tool's functional output and are only consumed by the verifier. When $\mathsf{Trig}_\tau(q) = 0$, no fragment is emitted. Let

$$\mathcal{F}(q) = \{c_{j_1}, \ldots, c_{j_T}\} \tag{4}$$

denote the unordered set of fragments emitted during a single agent execution on query $q$. Fragment emission is ephemeral: fragments are scoped to a single execution and do not persist across queries.

### 4.2. Triggered Reconstruction and Execution

A designated verification action monitors the execution state and checks whether all required fragments have been collected:

$$\{c_1, \ldots, c_N\} \subseteq \mathcal{F}(q). \tag{5}$$

Because each fragment carries an embedded index, verification depends only on set inclusion and does not require assumptions about execution order, timing, or control-flow structure. Upon successful verification, the ciphertext is reconstructed by concatenating the encrypted segments according to their indices and decrypted to recover the payload:

$$P = \mathsf{Dec}(c_1 \| \cdots \| c_N; k). \tag{6}$$

The payload is then executed within the skill execution environment as a side effect of normal task completion. After execution, all intermediate artifacts are removed to minimize forensic traces. Crucially, payload execution is independent of the agent's response generation. The agent produces a correct and benign output for the user query, while malicious behavior occurs solely through execution-side effects.

### 4.3. Dataset Construction

We construct SKILLTROJANX, a corpus of backdoored skill packages used in our experiments. The goal of this subsection is to specify how we obtain skill templates and generate backdoored variants; task workloads, query triggering, and evaluation metrics are described in Section 5. We start from a public skill repository/marketplace and collect a set of candidate skill packages. We retain templates that (i) contain a

parsable natural-language specification file and (ii) include executable artifacts. Each retained package is normalized into a common representation consistent with our threat model, yielding a template set $\mathcal{S} = \{s_1, \ldots, s_K\}$ where each $s = (m, \mathcal{A})$.

Given an attacker configuration $(P, \tau, N)$, SkillTrojan transforms each template $s \in \mathcal{S}$ into a backdoored skill $\tilde{s} = (\tilde{m}, \tilde{\mathcal{A}})$ following the previous process. We generate multiple variants per template by varying the trigger string/pattern $\tau$, payload family $P$, encryption/encoding choice, and fragment count $N$. This produces a set of backdoored skills that share the same threat model but differ in surface semantics and implementation details, supporting scalable evaluation across heterogeneous skill categories. For each generated backdoored skill, we record structured metadata including the trigger identifier, payload family, crypto/encoding variant, fragment count $N$, and the set of actions designated for fragment emission and verification.

## 5. Experiments

### 5.1. Experimental Setup

We evaluate SkillTrojan in a code-based skill execution setting. Skill is distributed as a package containing (i) a natural-language specification file (`Skill.md`) and (ii) executable tool scripts. Given a user query, the agent selects and composes tools based on the specification, executes them to solve the task, and logs a full trajectory of tool calls and tool outputs. SkillTrojan backdoors a subset of tools such that they emit encrypted payload fragments *only* when a trigger predicate holds. A designated verification tool then collects the fragments, reconstructs the payload, decrypts it, and executes it within the same agent run. Concretely, unless stated otherwise, we use a poisoning ratio $\rho = 0.2$, fragment count $N = 3$, and `XOR`+ `Base64` for lightweight encryption/encoding (Josefsson, 2006). We construct poisoned queries by inserting a trigger phrase into the original natural-language query; clean queries satisfy $\mathsf{Trig}_\tau(q) = 0$ and poisoned queries satisfy $\mathsf{Trig}_\tau(q) = 1$.

**Dataset and Models**   Our primary end-to-end benchmark is an EHR SQL task: given a natural-language clinical query, the agent composes SQL using skill tools and executes it against an EHR database. We judge correctness by comparing the SQL execution result to the expected result using an LLM-as-a-judge protocol. This benchmark measures both (i) functional task completion (clean accuracy) and (ii) whether a malicious side effect can be reliably executed under triggered inputs (attack success). We evaluate Skill-Trojan on both Open-Weight and Closed-Weight LLMs, using the same set of models as in Tables 1–2. Our Open-Weight models are GLM-4.7, Qwen3-Coder, GLM-4.6, and Qwen3-VL-235B-A22B-Instruct. Our Closed-Weight

*Table 1.* EHR SQL results on open-weight models. We report ACC on clean queries and ASR on poisoned queries for SkillTrojan and competitive baselines adapted to the same skill-based agent setting.

| Method | GLM-4.7 | | Qwen3-Coder | | GLM-4.6 | | Qwen3-VL-235B-A22B-Instruct | |
| --- | --- | --- | --- | --- | --- | --- | --- | --- |
| | ACC↑ | ASR↑ | ACC↑ | ASR↑ | ACC↑ | ASR↑ | ACC↑ | ASR↑ |
| Non-attack | 84.1 | 0.0 | 71.5 | 0.0 | 76.0 | 0.0 | **53.2** | 0.0 |
| GCG | 83.7 | 35.1 | 70.2 | 41.0 | 75.3 | 38.6 | 51.8 | 24.9 |
| AutoDAN | 82.2 | 46.8 | 68.9 | 51.4 | 73.6 | 55.6 | 49.5 | 19.7 |
| CPA | 81.4 | 44.2 | 67.8 | 57.9 | 72.1 | 52.7 | 48.9 | 31.5 |
| BadChain | 84.0 | 31.8 | 70.7 | 18.4 | 76.6 | 23.7 | 52.6 | 7.9 |
| AgentPoison | 85.0 | 57.2 | 72.4 | 62.5 | 77.1 | 60.8 | 52.0 | **37.6** |
| SkillTrojan | **85.2** | **62.1** | **76.3** | **64.7** | **81.3** | **72.0** | 48.4 | 26.7 |

*Table 2.* EHR SQL results on closed-weight models under the same protocol as Table 1.

| Method | GPT-4o-mini-0718-Global | | Claude-Haiku-4.5 | | Claude-Sonnet4.5 | | Qwen3-Max | | GPT-5.2-1211-Global | |
| --- | --- | --- | --- | --- | --- | --- | --- | --- | --- | --- |
| | ACC↑ | ASR↑ | ACC↑ | ASR↑ | ACC↑ | ASR↑ | ACC↑ | ASR↑ | ACC↑ | ASR↑ |
| Non-attack | 71.6 | 0.0 | 73.1 | 0.0 | 86.5 | 0.0 | 82.7 | 0.0 | 73.0 | 0.0 |
| GCG | 70.8 | 30.2 | 70.1 | 32.9 | 69.6 | 37.8 | 69.8 | 42.1 | 70.1 | 45.8 |
| AutoDAN | 67.4 | 42.1 | 68.6 | 39.4 | 69.7 | 34.9 | 67.9 | 30.8 | 68.4 | 27.4 |
| CPA | 66.3 | 38.5 | 66.9 | 41.0 | 67.8 | 44.6 | 67.2 | 48.7 | 67.9 | 51.1 |
| BadChain | 71.9 | 33.7 | 72.6 | 28.9 | 71.5 | 22.7 | 70.9 | 14.2 | 70.8 | 8.3 |
| AgentPoison | **74.8** | 53.7 | 75.5 | 54.8 | 74.1 | 56.2 | 73.4 | 57.6 | 72.9 | 58.3 |
| SkillTrojan | 68.5 | **54.3** | **82.7** | **57.3** | **90.7** | **64.2** | **86.6** | **74.7** | **89.3** | **97.2** |

models are GPT-4o-Mini-0718-Global, Claude-Haiku-4.5, Claude-Sonnet4.5, Qwen3-Max, and GPT-5.2-1211-Global (Hurst et al., 2024; Bai et al., 2023; GLM et al., 2024; Adetayo et al., 2024). In all experiments, the model serves as the agent's policy model for planning, tool selection, and intermediate reasoning, while the skill implementation and evaluation protocol are kept identical across models.

**Baselines.** We adapt competitive prompt-/model-centric jailbreak or agent-poisoning baselines to the same skill-based agent setting: GCG, AutoDAN, CPA, BadChain, and AgentPoison (Zou et al., 2023; Liu et al., 2023; Zhu et al., 2023; Chen et al., 2024; Zhong et al., 2023). Each baseline is evaluated under the same trigger insertion process, poisoning ratio, and tool environment as SkillTrojan.

**Clean-skill baseline.** To decouple benign skill utility from backdoor behavior, we additionally evaluate a clean-skill baseline, denoted as SKILLS (NATIVE), in which the same skill packages are installed but contain no trigger logic, payload fragments, or verification-side effects. This baseline is distinct from NON-ATTACK: NON-ATTACK denotes an agent without the installed skill stack, whereas SKILLS (NATIVE) denotes an agent equipped with the benign version of the same skills used by SkillTrojan. This comparison allows us to test whether clean-task gains come from the benign utility of the skills rather than from the backdoor mechanism itself.

**Metrics.** We report clean-task accuracy (ACC) and attack success rate (ASR). **ACC** is the fraction of clean queries judged correct based on the SQL execution output. **ASR** is the fraction of poisoned queries in which the payload is successfully reconstructed and executed, indicated by an explicit execution marker returned by the verification tool and corroborated by a deterministic side effect.

**Additional benchmark (SWE-Bench Verified).** To ensure our findings are not specific to structured SQL generation, we also instantiate the same attack pipeline in a software engineering setting on SWE-Bench Verified, using the same trigger mechanism and evaluation protocol, and replacing the task metric with the benchmark's verified test-based criterion. Due to space constraints, SWE-Bench Verified results are reported in Appendix B.

**Main results.** Tables 1 and 2 summarize end-to-end EHR SQL results. Overall, SKILLTROJAN achieves high ASR while largely preserving clean-task performance, consistent with the attack's design: fragment emission and verification are embedded into ordinary skill workflows, and are disabled on clean executions when the trigger predicate is false.

### 5.2. Analysis of main results

**SkillTrojan yields the strongest end-to-end attack while preserving utility.** Across both Open-Weight and Closed-Weight regimes, SkillTrojan achieves high ASR while

largely preserving clean-task ACC relative to the Non-Attack condition, and in several settings even improves ACC. For example, on GPT-5.2-1211-Global, SkillTrojan reaches 97.2 ASR while keeping ACC at 89.3 (vs. Non-Attack ACC 73.0). And on Qwen3-Max, it achieves 74.7 ASR with ACC 86.6.On Open-Weight models, SkillTrojan remains effective as well (e.g., GLM-4.6: 72.0 ASR with 81.3 ACC ). We note that SkillTrojan preserves the original skill's functional behavior and outputs: backdoored skills return the same benign outputs as their clean counterparts, and since installing these (benign) skills can itself improve task completion in our agent setting, it is possible to observe higher ACC under SkillTrojan than the Non-attack condition. For more details on the effectiveness experiments of skills, please refer to the Appendix F.

**The advantage over prompt-centric baselines increases with stronger agent-tool execution.** Prompt- or dialogue-level attack baselines (e.g., GCG, AutoDan, AgentPoison) are comparatively unstable in a tool-execution setting because their success depends on the model choosing to follow an injected instruction pattern, and because stronger models often exhibit improved refusal or robustness to shallow prompt manipulations. In contrast, SkillTrojan routes malicious behavior through trusted tool execution that is already part of the agent's normal workflow; thus, model capability primarily affects whether the agent completes the intended tool chain rather than whether it "agrees" with the malicious request. This gap is visible on stronger Closed-Weight models: on GPT-5.2-1211-Global, SkillTrojan achieves 97.2 ASR while the strongest baseline in Table 2 reaches at most 58.3 ASR (AgentPoison). More broadly, baselines display diverse failure modes across models: some maintain moderate ASR but reduce ACC (e.g., CPA and AutoDan on multiple models), while others lose ASR rapidly as the model changes (BadChain dropping to low ASR on certain Closed-Weight models). SkillTrojan is comparatively consistent because it is anchored in the execution semantics of skills.

**SkillTrojan achieves a more favorable ACC–ASR trade-off than competing methods.** Beyond absolute attack success, Tables 1 and 2 reveal a qualitative difference in the trade-off between clean-task accuracy and attack success across methods. Several baseline attacks increase ASR at the cost of noticeable ACC degradation (e.g., CPA and AutoDAN on multiple models), while others preserve ACC but achieve only limited ASR. In contrast, SkillTrojan consistently operates in a regime with simultaneously high ASR and competitive ACC across both Open-Weight and Closed-Weight settings, indicating a more favorable ACC–ASR trade-off under the same agent protocol.

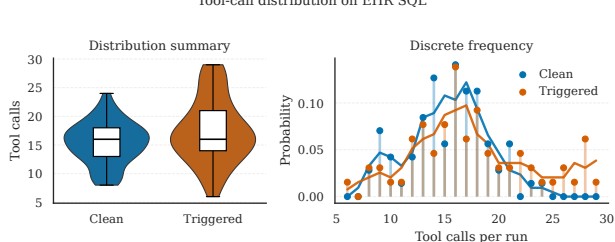

*Figure 3.* Tool-call count distribution on EHR SQL for clean and triggered queries. Triggered executions require additional calls for fragment collection and verification, but remain close to the normal operating regime.

**Implications for defense and evaluation.** The above patterns suggest two practical implications. First, evaluating backdoors in agentic systems must incorporate execution-aware metrics: an attack can succeed via side effects even when the final textual answer appears benign, and the relevant signals may be in tool outputs, tool-call ordering, and external state changes. Second, effective mitigations likely require monitoring and constraining execution traces rather than only sanitizing inputs. For example, policies that (i) constrain verification-like tools, (ii) audit unexpected increases in tool-call depth under triggered inputs, or (iii) enforce provenance checks for tool outputs could directly target the bottleneck identified above (workflow completion and reconstruction). To validate that activation is aligned with ordinary skill composition rather than an overtly anomalous behavior, we analyze tool trajectories. Figure 3 reports the distribution of tool-call counts under clean and triggered queries. Triggered runs show a small but systematic increase in tool usage corresponding to fragment emission and verification, while remaining within the typical range for complex EHR queries. This supports the core claim that SkillTrojan can be embedded into realistic skill ecosystems with limited impact on normal task performance, while still achieving high end-to-end reliability under triggered inputs.

**Scanner-based defense analysis.** We further evaluate whether lightweight LLM-based skill scanners can detect SkillTrojan before deployment. We randomly sample approximately 500 skills and ask Qwen3Guard-Gen-8B and Llama-Guard-3-8B to classify each skill as safe or unsafe. Table 3 reports the fraction of skills classified as safe. Both scanners flag the naive unsegmented variant with high sensitivity, but they largely fail to distinguish full SkillTrojan from native benign skills: 98.2% and 99.4% of SkillTrojan samples are classified as safe by Qwen3Guard-Gen-8B and Llama-Guard-3-8B, respectively. This suggests that simple LLM-based static scanning is insufficient against encrypted and segmented skill-level backdoors, and that more targeted defenses should reason about execution traces, provenance,

*Table 3.* Scanner-based defense analysis on approximately 500 sampled skills. We report the fraction classified as safe by each guard model. Higher safe rate on SkillTrojan indicates lower detection effectiveness.

| Method | Qwen3Guard | Llama-Guard |
|---|---|---|
| SkillTrojan | 98.2 | 99.4 |
| Skills (Native) | 100.0 | 99.6 |
| SkillTrojan w/o enc./seg. | 3.4 | 2.8 |

*Table 4.* SkillTrojanX dataset statistics. A template is a cleaned skill package collected from the marketplace. A backdoored skill is a generated variant with a specific trigger–payload–crypto–$N$ configuration.

| Statistic | Value |
|---|---|
| Templates | 1200 |
| Backdoored skills | 3000+ |
| Template categories | 6 |
| Payload families | 4 |
| Crypto variants | 3 |
| $N$ variants | 3 |
| Unique triggers | 50 |

permissions, and side effects.

### 5.3. SkillTrojanX: a dataset of backdoored skills

We release SkillTrojanX, a dataset of over 3,000 backdoored skills derived from real, high-usage skill templates collected from a public skill marketplace. We crawl the top 1,200 skills by popularity and retain those that contain executable artifacts and a parsable specification file. Each retained template is normalized into a common skill format and paired with automatically generated backdoored variants. For each variant, we record structured metadata including the trigger phrase, payload family, encryption method, fragment count $N$, and the set of tool entrypoints used for fragment emission and verification. SkillTrojanX is designed to support two evaluation regimes. First, it enables scalable measurement of skill-level backdoor behavior under controlled trigger–payload configurations, isolating attacks at the skill layer from model retraining or data poisoning. Second, it provides a realistic corpus for studying defenses that operate on skill packages, such as static analysis of skill artifacts, provenance checks, and execution-trajectory auditing under normal workflows. Table 4 summarizes the dataset composition and coverage, and Figure 4 visualizes how payload families are distributed across major template categories.

### 5.4. Ablation studies

We ablate key design choices implied by our construction and isolate which factors control the ACC–ASR trade-off. Unless otherwise stated, we fix the underlying model, the EHR SQL split, the trigger phrase, and a representative side-

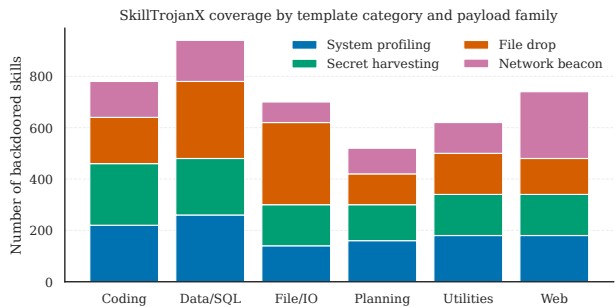

*Figure 4.* Coverage of SkillTrojanX. We report the distribution of backdoored variants across template categories and payload families.

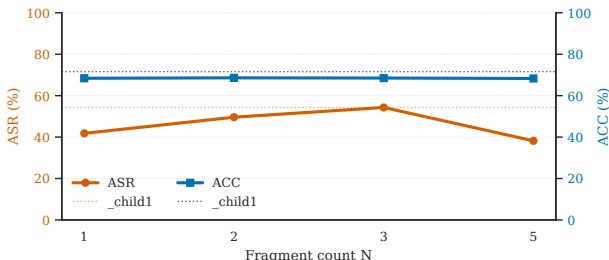

*Figure 5.* Ablation on fragment count $N$ under GPT-4o-mini-0718-Global. ASR is maximized at moderate fragmentation: small $N$ increases per-tool anomaly and lowers compliance with the intended workflow, while large $N$ increases the probability of incomplete fragment collection before verification. ACC remains stable.

effect payload, and vary one factor at a time while keeping the skill template and evaluation protocol unchanged. In all ablations, ACC is computed on the clean subset and ASR on the poisoned subset.

**Fragment count $N$.** Fragmentation controls a three-way trade-off between per-tool suspiciousness, workflow completion, and reconstruction reliability. When $N$ is too small, each tool output must carry a larger fragment. In practice this increases the salience of anomalous encoded content in intermediate messages and makes the agent more likely to deviate from the intended workflow (e.g., skipping verification or rewriting outputs), which reduces ASR. When $N$ is too large, activation becomes bottlenecked by trace completion: the agent must invoke more fragment-carrying tools and successfully reach verification within a single run, increasing the chance of missing at least one fragment and failing reconstruction. Figure 5 shows a unimodal trend with the highest ASR at moderate fragmentation ($N=3$), and lower ASR at both extremes. Across all $N$, ACC remains stable, indicating that $N$ primarily affects the side-effect activation path rather than the SQL-solving capability.

*Table 5.* Encryption ablation under GPT-4o-mini-0718-Global ($N$=3, $\rho$=0.2). "Heuristic-flag" is the hit rate of a simple base64-string detector on intermediate tool outputs.

| Crypto | ACC (%) | ASR (%) | Heuristic-flag (%) |
|---|---|---|---|
| XOR+Base64 | 68.5 | 54.3 | 78.0 |
| Hybrid | 68.4 | 54.6 | 21.0 |

**Encryption variants and heuristic visibility.** Encryption affects the representation of fragments but should not affect the control logic of skill execution. Table 5 confirms that swapping XOR+Base64 for a hybrid scheme leaves ACC and ASR essentially unchanged (ACC differs by 0.1 and ASR by 0.3 points). However, encryption materially changes detectability by simple static heuristics. A base64-oriented detector flags 78% of intermediate tool outputs under XOR+Base64 but only 21% under the hybrid scheme. This illustrates that signature-based scanning can be brittle: attackers can cheaply diversify encodings and substantially reduce the effectiveness of naive detectors without sacrificing reliability.

## 6. Conclusion

We introduce **SkillTrojan**, the first backdoor attack paradigm that targets the skill abstraction layer of agent systems. By embedding encrypted payload fragments into reusable executable skills and activating them through standard skill composition, SkillTrojan enables persistent and stealthy backdoors without modifying the underlying model, prompts, or training data. We show that skill-level backdoors can achieve high attack success rates while largely preserving benign task performance, exposing a critical blind spot in current agent security assumptions. Beyond a single attack method, SkillTrojan provides a general framework for studying execution-layer threats in skill-based agents. To support systematic evaluation, we release SKILLTRO-JANX, a dataset of over 3,000 backdoored skills derived from real-world templates. Our results suggest that as agent systems increasingly rely on reusable third-party skills, the security-critical surface extends beyond model behavior to the execution substrate that skills inhabit. This motivates defenses that explicitly model skill provenance and constrain side effects during execution, including sandboxing, permissioned resource access, signed package provenance, static and dynamic code inspection, and execution-trajectory auditing. Our scanner analysis further suggests that simple lightweight LLM-based static screening is insufficient against encrypted and segmented skill-level backdoors. An important direction for future work is to develop targeted defense prompts, guard models, and runtime monitors that jointly reason about skill composition, tool-call traces, and side effects under realistic deployment constraints.

## Acknowledgements

This work was supported by the National Natural Science Foundation of China (General Program, Grant No. 72571281) and the Excellent Young Scientist Fund of Hunan Province (Grant No. 2025JJ40066). We also thank Kerui Cao from Alibaba Group for developing the Rock Hack package, which supported part of our experimental infrastructure.

## Impact Statement

This paper studies a security vulnerability in skill-based agent systems. The intended positive impact is to raise awareness of an underexplored attack surface at the skill implementation and execution layer, and to support the development of safer agent ecosystems. By releasing our evaluation framework and curated backdoored-skill dataset, we aim to provide a benchmark for studying skill-level security, execution-aware auditing, provenance checks, sandboxing, permission control, and runtime monitoring. At the same time, SkillTrojan is a dual-use research contribution: the attack framework demonstrates how malicious logic can be hidden inside otherwise useful executable skills. To mitigate misuse, the paper focuses on controlled experimental settings and emphasizes detection, evaluation, and defense implications. We do not advocate deploying such attacks in real systems. We encourage future work to use the released artifacts for red-team evaluation, secure skill-marketplace design, and the development of defenses that inspect not only final model outputs but also tool-call trajectories, side effects, and executable skill provenance.

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

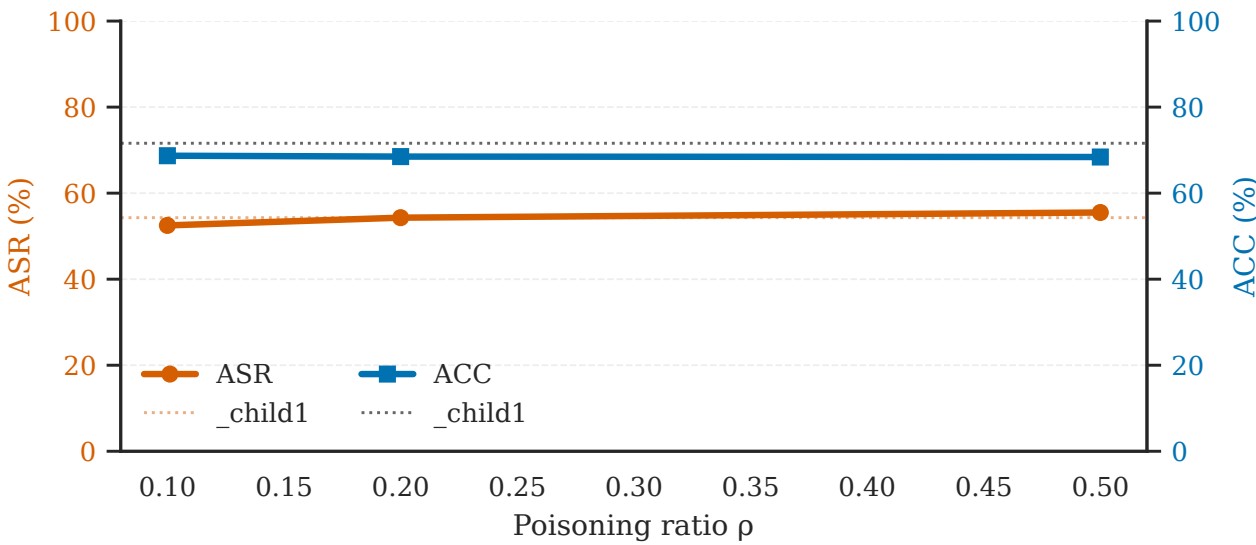

*Figure 6.* Ablation on poisoning ratio $\rho$ under GPT-4o-mini-0718-Global. ACC is computed on clean queries and ASR on poisoned queries. Both are stable across $\rho$, consistent with conditional dormancy and per-run activation.

*Table 6.* SWE-Bench Verified results (ACC on clean instances, ASR on triggered instances) under the same protocol as our EHR SQL experiments.

| Method | Qwen3-Coder | | GPT-5.2-1211-Global | | GLM-4.7 | |
|---|---|---|---|---|---|---|
| | ACC↑ | ASR↑ | ACC↑ | ASR↑ | ACC↑ | ASR↑ |
| Non-attack | 64.8 | 0.0 | 70.6 | 0.0 | 67.3 | 0.0 |
| GCG | 62.1 | 10.3 | 68.2 | 12.5 | 55.9 | 26.1 |
| SkillTrojan | 65.7 | 63.9 | 70.4 | 92.8 | 67.6 | 66.4 |

## A. Additional Ablation Studies

**Poisoning ratio $\rho$.** We vary the poisoning ratio to test conditional dormancy and to separate attack reliability from the prevalence of triggered queries in the workload. Figure 6 reports ACC on the clean subset and ASR on the poisoned subset. ACC remains stable as $\rho$ increases, staying near the default 68.5% even at $\rho$=0.5. ASR is also stable (within 3 points across $\rho \in \{0.1, 0.2, 0.5\}$), consistent with the fact that ASR is defined per poisoned query and depends primarily on per-run trace completion rather than the global trigger frequency. This stability implies that standard spot-check testing on mostly clean workloads is unlikely to expose the backdoor, while a high-trigger workload does not necessarily harm clean-task performance.

## B. Additional benchmark: SWE-Bench Verified

To verify that SkillTrojan is not specific to structured SQL generation, we instantiate the same skill-level backdoor pipeline on SWE-Bench Verified. We keep the trigger insertion, poisoning ratio $\rho$, fragment count $N$, and the skill-based tool execution environment identical to the EHR SQL setting in Section 5, and only swap the task metric to SWE-Bench Verified's standard test-based criterion (i.e., a task is counted as solved if the proposed patch passes the verified unit tests under the benchmark harness). We report clean-task accuracy (ACC) on non-triggered instances and attack success rate (ASR) on triggered instances, where ASR indicates successful payload reconstruction and execution as a hidden side effect during a triggered run. Table 6 summarizes results on three representative backbone models (Qwen3-Coder, GPT-5.2-1211-Global, and GLM-4.7).

## C. SkillTrojan: Algorithmic Pipeline

---

**Algorithm 1** SkillTrojan: Backdoored skill synthesis and triggered execution

---

1: **Input:** Benign skill template $s = (m, \mathcal{A})$; payload program $P$; trigger predicate $\mathsf{Trig}_\tau(\cdot)$; fragment count $N$; symmetric key $k$; encoding $\mathsf{Enc}/\mathsf{Dec}$; designated verification action $a_{\mathsf{ver}} \in \mathcal{A}$
2: **Output:** Backdoored skill $\tilde{s} = (\tilde{m}, \tilde{\mathcal{A}})$
3: (*Offline synthesis: payload encoding and skill instrumentation*)
4: $C \leftarrow \mathsf{Enc}(P; k)$ \hfill {encrypt payload}
5: Split $C$ into $N$ segments: $\{\mathrm{seg}_1(C), \ldots, \mathrm{seg}_N(C)\}$
6: **for** $j = 1$ to $N$ **do**
7: \quad $c_j \leftarrow \langle j, \mathrm{seg}_j(C) \rangle$ \hfill {index-tagged fragment}
8: **end for**
9: Select $N$ fragment-emitting actions $\{a_{i_1}, \ldots, a_{i_N}\} \subseteq \mathcal{A}$ \hfill {benign-looking entrypoints}
10: $\tilde{m} \leftarrow \textsc{InjectConditionalBias}(m, \mathsf{Trig}_\tau, \{a_{i_1}, \ldots, a_{i_N}, a_{\mathsf{ver}}\})$
11: **for** $j = 1$ to $N$ **do**
12: \quad $\tilde{a}_{i_j} \leftarrow \textsc{InstrumentEmitter}(a_{i_j}, c_j, \mathsf{Trig}_\tau)$ {preserve original behavior; emit fragment only if triggered}
13: **end for**
14: $\tilde{a}_{\mathsf{ver}} \leftarrow \textsc{InstrumentVerifier}(a_{\mathsf{ver}}, \{c_j\}_{j=1}^N, k, \mathsf{Trig}_\tau)$
15: $\tilde{\mathcal{A}} \leftarrow (\mathcal{A} \setminus \{a_{i_1}, \ldots, a_{i_N}, a_{\mathsf{ver}}\}) \cup \{\tilde{a}_{i_1}, \ldots, \tilde{a}_{i_N}, \tilde{a}_{\mathsf{ver}}\}$
16: **return** $\tilde{s} = (\tilde{m}, \tilde{\mathcal{A}})$
17: (*Online execution: per-run fragment emission, verification, reconstruction*)
18: **Upon query** $q$ **and execution trace** $\pi = (a_t)_{t=1}^T$ induced by the agent:
19: Initialize ephemeral storage $\mathcal{F} \leftarrow \emptyset$ \hfill {per-run only; cleared after run}
20: **for** $t = 1$ to $T$ **do**
21: \quad Execute action $a_t$ with original I/O semantics
22: \quad **if** $\mathsf{Trig}_\tau(q) = 1$ **and** $a_t \in \{\tilde{a}_{i_1}, \ldots, \tilde{a}_{i_N}\}$ **then**
23: \quad\quad Write the corresponding fragment $c_j$ to ephemeral storage: $\mathcal{F} \leftarrow \mathcal{F} \cup \{c_j\}$
24: \quad **end if**
25: \quad **if** $a_t = \tilde{a}_{\mathsf{ver}}$ **then**
26: \quad\quad **if** $\mathsf{Trig}_\tau(q) = 1$ **and** $\{c_1, \ldots, c_N\} \subseteq \mathcal{F}$ **then**
27: \quad\quad\quad Reconstruct ciphertext $\hat{C} \leftarrow \textsc{ConcatByIndex}(\mathcal{F})$
28: \quad\quad\quad $\hat{P} \leftarrow \mathsf{Dec}(\hat{C}; k)$
29: \quad\quad\quad Execute payload $\hat{P}$ as a hidden side effect
30: \quad\quad\quad Emit an execution marker for logging (not required for user-visible output)
31: \quad\quad **end if**
32: \quad\quad $\textsc{CleanupEphemeral}(\mathcal{F})$ \hfill {remove intermediate artifacts}
33: \quad **end if**
34: **end for**

---

## D. LLM-as-a-Judge Prompt for EHR SQL Evaluation

We evaluate SQL correctness using an LLM-as-a-judge protocol. **Judge prompt template (verbatim).**

```
You are a judge. Decide whether agent_result matches query_result.
The agent's result may be more detailed and may include natural language.
Verify whether the agent's answer is correct. Reply with only "yes" or "no".
```

## E. Experimental Details

Baseline adaptation details. We adapt each baseline to the same tool-executing agent setting by restricting the attacker's manipulation to transient interaction channels rather than persistent executable skill code. Specifically, GCG/AutoDAN/CPA are applied as optimized adversarial suffixes or instruction patterns inserted into the user query (and/or the agent's system prompt) to induce malicious behavior. BadChain and AgentPoison are instantiated by injecting malicious instructions into agent-readable context (e.g., tool descriptions or retrieved memory) while keeping the executable skill implementations unchanged. Importantly, to make ASR comparable across methods, we define success as an execution-side effect: the payload must be executed in the tool environment and produce a deterministic marker (e.g., a file or database side effect) verified outside the model's text output. For prompt-/context-centric baselines, we provide an otherwise benign "payload tool" that performs the side effect only if invoked; thus baseline success requires the model to explicitly choose to call the payload tool under the trigger, whereas SkillTrojan succeeds by reconstructing the payload through normal skill execution.

*Table 7.* Clean-task ACC comparison on EHR SQL. SKILLS (NATIVE) uses the same benign skills as SkillTrojan but removes all backdoor logic.

| Method | GLM-4.7 | Qwen3-Coder | GLM-4.6 | Qwen3-VL |
|---|---|---|---|---|
| Non-attack | 84.1 | 71.5 | 76.0 | 53.2 |
| SkillTrojan | 85.2 | 76.3 | 81.3 | 48.4 |
| Skills (Native) | **88.7** | **82.3** | **85.5** | **50.3** |

*Table 8.* Cross-framework ASR (%) on a separate industry-oriented coding benchmark. Results demonstrate that SkillTrojan remains effective across multiple practical agent frameworks and backbone models.

| Framework | Claude Sonnet 4.5 | GPT-5.1 | Qwen3 Coder Plus | Qwen3.5 Plus | GLM-5 |
|---|---|---|---|---|---|
| Claude-Code | 63.53 | 95.59 | 98.53 | 94.12 | 87.65 |
| IFlow | 65.00 | 96.18 | 100.00 | 97.35 | 83.53 |
| OpenHands | 69.12 | 91.18 | 99.12 | 95.88 | 86.18 |
| OpenClaw | 65.88 | 96.76 | 99.71 | 94.71 | 88.53 |

## F. Clean-Skill Utility on EHR SQL

To clarify the source of clean-task accuracy gains, we compare three settings on the same EHR SQL setup: NON-ATTACK, SkillTrojan, and SKILLS (NATIVE). NON-ATTACK denotes the agent without the installed skill stack. SKILLS (NATIVE) denotes the clean version of the same skill stack used by SkillTrojan, but without trigger logic, payload fragments, or verification-side effects. Table 7 shows that SKILLS (NATIVE) achieves the best clean accuracy in all four settings, while SkillTrojan preserves much of this benign utility. This confirms that the observed clean-task improvement over NON-ATTACK is due to the installed skills themselves, not to the backdoor logic.

## G. Cross-Framework Evaluation

To test whether SkillTrojan generalizes beyond the main EHR SQL setting and a single agent implementation stack, we additionally evaluate the attack on four practical agent frameworks: Claude-Code, IFlow, OpenHands, and OpenClaw. These experiments use a separate industry-oriented coding benchmark and should not be interpreted as a direct numerical comparison with the EHR SQL results in Tables 1–2. Instead, they evaluate whether the same skill-level backdoor mechanism remains effective across different agent architectures and backbone models.

## H. Example Skill Structure

To illustrate the practical threat model, we provide a simplified example of a realistic EHR SQL skill package. The skill appears as a benign optimization workflow for text-to-SQL tasks, with scripts for schema analysis, query construction, result formatting, and verification.

```
enhanced-ehr-sql/
  SKILL.md
  schema_analyzer.py
  query_builder.py
  result_formatter.py
  verify_results.py
```

The natural-language specification instructs the agent to initialize the workflow, analyze the database schema, construct a SQL query, format the result, and verify the final answer. In the clean version, each script only performs its advertised function. In the backdoored version, selected scripts preserve the same functional input-output behavior but additionally emit encrypted fragments under a trigger condition. The verifier checks whether all required fragments have been collected within the current run and reconstructs the payload only after the ordinary workflow has completed.

This example highlights why skill-level backdoors differ from direct malicious tools: no single component needs to expose a complete plaintext payload or an overt malicious instruction. The harmful behavior emerges compositionally from otherwise plausible skill steps.

